evolution, palaeontology

species diversification, speciation, extinction, multivariate birth–death model, time-varying diversification, diversity-dependence

**Author for correspondence:**
Thomas A. Neubauer
e-mail: tneub@zo.jlug.de

# Drivers of diversification in freshwater gastropods vary over deep time

Thomas A. Neubauer[1,2], Torsten Hauffe[3], Daniele Silvestro[3,4], Christopher R. Scotese[5], Björn Stelbrink[1], Christian Albrecht[1], Diana Delicado[1], Mathias Harzhauser[6] and Thomas Wilke[1]

[1]Department of Animal Ecology and Systematics, Justus Liebig University, Heinrich-Buff-Ring 26-32 (IFZ), 35392 Giessen, Germany
[2]Marine Biodiversity, Naturalis Biodiversity Center, Darwinweg 2, 2333 CR Leiden, The Netherlands
[3]Department of Biology, University of Fribourg and Swiss Institute of Bioinformatics, Chemin du Musée 10, 1700 Fribourg, Switzerland
[4]Gothenburg Global Biodiversity Centre, University of Gothenburg, Carl Skottsbergs gata 22B, 41319 Gothenburg, Sweden
[5]Department of Earth and Planetary Sciences, Northwestern University, 2145 Sheridan Road, Evanston, IL 60208, USA
[6]Geological-Paleontological Department, Natural History Museum Vienna, Burgring 7, 1010 Vienna, Austria

TAN, 0000-0002-1398-9941; TH, 0000-0001-5711-9457; DS, 0000-0003-0100-0961; CRS, 0000-0002-9742-3581; BS, 0000-0002-7471-4992; CA, 0000-0002-1490-1825; DD, 0000-0002-6832-4170; MH, 0000-0002-4471-6655; TW, 0000-0001-8263-7758

Unravelling the drivers of species diversification through geological time is of crucial importance for our understanding of long-term evolutionary processes. Numerous studies have proposed different sets of biotic and abiotic controls of speciation and extinction rates, but typically they were inferred for a single, long geological time frame. However, whether the impact of biotic and abiotic controls on diversification changes over time is poorly understood. Here, we use a large fossil dataset, a multivariate birth–death model and a comprehensive set of biotic and abiotic predictors, including a new index to quantify tectonic complexity, to estimate the drivers of diversification for European freshwater gastropods over the past 100 Myr. The effects of these factors on origination and extinction are estimated across the entire time frame as well as within sequential time windows of 20 Myr each. Our results find support for temporal heterogeneity in the factors associated with changes in diversification rates. While the factors impacting speciation and extinction rates vary considerably over time, diversity-dependence and topography are consistently important. Our study highlights that a high level of heterogeneity in diversification rates is best captured by incorporating time-varying effects of biotic and abiotic factors.

## 1. Introduction

A major question in evolutionary biology concerns the processes that drive species diversification [1–9]. A variety of biotic and abiotic factors have been shown to explain variation in speciation and extinction rates. Biotic factors include diversity-dependence (i.e. higher species numbers slow down diversification, both within the focal species group and among clades [2,7,10–13]), as well as predatory, trophic or mutualistic interactions [14,15]. Abiotic factors proposed to affect diversification involve environmental features, such as climate change, sea-level fluctuations or continental fragmentation [1,4,16–20]. In some cases, a combined effect of abiotic and biotic factors has been demonstrated [3,4,17,21]. In geographically restricted regions, such as oceanic islands, ecosystem age, area and isolation also play an important role by constraining ecological opportunity [18,22,23].

The abundance of potential controls on diversification raises the fundamental question of whether diverging results across studies are due to differences in the analysed ecosystems and species group, the geographical scale (individual systems versus larger regions or global assessments), the applied methodology or a combination of these. Particularly when considering long time frames, drivers of diversification may change over time [24]. Several studies have addressed diversification from an extended temporal perspective (i.e. over tens to hundreds of millions of years) [2–4,16,25–27]. These studies used models that assume that biotic or abiotic factors have a fixed effect on diversification at any moment in time. Assuming that the same processes operate uniformly across such long time scales might severely hamper our ability to understand the factors controlling diversification by masking short-term effects. For example, geological processes like continental fragmentation vary at a slower pace than biotic interactions, and could therefore mask their effects in studies across an extended geological interval.

Most of our understanding about the controls of diversification derives from marine groups, mammals, plants and terrestrial insects [4,8,17,26–29]. Comparatively little is known about non-marine invertebrates other than insects. Even less is understood about the changing impact of diversification drivers over time, and the available knowledge is based on incompatible methodological approaches (fossils versus phylogenies, single clades versus entire faunas, empirical data versus simulations), geographical scales (single ecosystem versus global) and species groups. By analysing global family-level phylogenies of vertebrates and diatoms, Lewitus & Morlon [30] and Lewitus *et al.* [8] showed that a clade runs through different phases and patterns of diversification that are controlled by different evolutionary processes. They suggested that a clade's diversification is influenced by a time-sensitive series of biotic/abiotic factors. Based on simulations, Aguilée *et al.* [7] found that the influence of biotic processes (resource competition, genetic differentiation) and abiotic factors (landscape fluctuation) on speciation and extinction rates changes over time. Wilke *et al.* [13] came to a similar conclusion based on a 1.36 Myr fossil record of lacustrine diatoms, showing that both biotic and abiotic factors affected speciation in the early lake phase, while diversity-dependence and ecological limits controlled the speciation rate over most of the remaining study interval.

Here, we investigate potential variation in the influence of abiotic and biotic drivers of diversification over geologically extended time scales using an extensive fossil dataset of freshwater gastropod species. The choice of the taxonomic level may severely affect the outcome of such an investigation [14]. Analyses spanning long geological time frames have been performed primarily at the genus or family level or based on restricted species groups. However, conclusions on the processes driving diversification are shown to differ among taxonomic levels [31,32]. In order to provide reasonable estimates of diversification rates and their drivers, an ideal model taxon should have: (i) a high species-level diversity containing a sufficient number of speciation and extinction events to be evaluated; (ii) a wide geographical distribution, allowing general inferences of global and regional patterns; (iii) a good preservation and continuous fossil record over long geological periods, enabling analyses of variation in diversification rates and associated drivers;

(iv) a representative coverage of the habitat types and lifestyles (e.g. in feeding or reproduction) of the taxon in the fossil record, minimizing an ecological bias of the results; and (v) a robust taxonomic sampling and assessment [14].

The fossil record of European freshwater gastropods fulfils these criteria. By contrast to most other geographical regions of the world, the European record is well documented by hundreds of publications, which are considered to provide a representative image of past diversity [33,34]. In particular, the fossil record of the last 100 Myr offers sufficient coverage without major stratigraphic gaps [34]. Moreover, freshwater gastropods encompass several lifestyles [35] and include competing taxa [36] required to address potential diversity-dependent diversification limitations.

Combining freshwater gastropods as a highly diverse biota in a clearly defined continental setting and a set of biotic and abiotic factors, we provide new perspectives on the drivers of species diversification and their variation through time. Our set of predictor variables contains one biotic (diversity) and nine abiotic predictors: mean annual temperature, annual precipitation, continental area, mean elevation range, average basin size and number of basins, terrain ruggedness index (TRI), mean geographical distance among fossil occurrences and a novel tectonic complexity index (TCI). We hypothesize that the factors driving diversification in European freshwater gastropods vary over time and test that by assessing the influence of abiotic/biotic factors on diversification rates across multiple time frames (hereafter referred to as 'time windows'). We compare the outcomes of the analyses based on a single long time frame versus several shorter time windows to evaluate the potential advantage of applying a time-varying approach. We also evaluate how the influence of biotic and abiotic factors varies across time windows to assess specifically the relative roles of abiotic and biotic factors for the diversification of freshwater gastropods.

## 2. Material and methods

### (a) Fossil occurrence data and diversification rates

The underlying data and diversification rate analyses are based on Neubauer *et al.* [34]. Origination and extinction times were inferred for 3122 species based on a dataset of 24 759 Jurassic–Pleistocene occurrences of European freshwater gastropods (5564 localities; electronic supplementary material, figure S1.1) using the open-source program PyRate v. 3 [2,37,38]. We used the birth–death model with shifts (BDS; i.e. speciation and extinction rates are allowed to shift across time frames), with the number of shifts and their ages inferred by a reversible jump Markov chain Monte Carlo (RJ-MCMC) algorithm [38] (see electronic supplementary material for details).

### (b) Multivariate birth–death analyses

The potential influence of selected biotic and abiotic factors on speciation and extinction rates was tested with the multivariate birth–death (MBD) model developed by Lehtonen *et al.* [4]. The influence of biotic or abiotic factors on speciation/extinction rates was assessed via shrinkage weights, where mean values across all replicates exceeding 0.5 indicate significant support for the corresponding parameter [4]. Exponential links between predictors and rates were chosen since this method better accommodates the fact that rates cannot be negative [4]. All predictor data were interpolated to 0.1 Myr steps, $\log_{10}$-transformed and

**Table 1.** Predictors used in the multivariate birth–death (MBD) model. Indicated are the type of factor and information on how it was inferred. DEM, digital elevation model.

| predictor | type | inferred based on |
| --- | --- | --- |
| continental area | geographic | palaeo-DEMs |
| number of basins | topographic | palaeo-DEMs |
| basin size | topographic | palaeo-DEMs[a] |
| diversity | biotic | birth–death model with shifts |
| elevation | topographic | palaeo-DEMs[a] |
| geographical distance | geographic | palaeocoordinates |
| tectonic complexity index | topographic/geographic | palaeo-DEMs |
| annual precipitation | climatic | palaeo-climate data[a] |
| mean annual temperature | climatic | palaeo-climate data[a] |
| topographic ruggedness index | topographic | palaeo-DEMs[a] |

[a]Predictors that are based on the mean of the values measured at the palaeocoordinates of fossil localities per time slice; see Material and methods and electronic supplementary material for details.

standardized to zero mean and a standard deviation of one to facilitate comparison.

Considering the large uncertainties and limited temporal variation in the estimated rates for the Jurassic and Early Cretaceous [34], we restricted the MBD analyses to the last 100 Myr. As drivers of diversification may change over time, traditional approaches that link proxies and rates over extended time scales may fail to detect significant relationships on shorter scales. We ran test analyses to assess a reasonable temporal duration of the time windows, in which the complete time interval (0–100 Myr) can be segmented. However, we had to limit our window selection due to data resolution and computational reasons. The complete time interval was dissected into five 20 Myr windows. Preliminary tests using ten 10 Myr windows resulted repeatedly in several runs without parameter convergence despite a high number of MCMC generations.

For each time window, the final MBD analyses were run on 100 replicates (20 Myr windows) and 25 replicates (100 Myr window) of speciation and extinction times, respectively. The initial settings were 50 million (20 Myr windows) and 100 million MCMC generations (100 Myr window), sampling every 50 000, omitting the first 10 million (20 Myr windows) and 20 million generations (100 Myr window) as burn-in. The analyses for some replicates of the 20 Myr windows yielded low effective sampling sizes (less than 100) and were repeated with 150 million or 300 million MCMC generations (at the same sampling frequency). Only a few runs (26 out of 525 runs in total) were returned with still low effective sampling sizes even at a high number of MCMC generations. To avoid biasing the results, we excluded runs with effective sampling size for posterior, prior or likelihood less than 100.

We evaluated whether smaller time windows offer an improvement over the entire time frame by quantifying the mean absolute percentage error (MAPE; compare Andermann *et al.* [39]) of the reconstructed MBD rates in comparison to the rates inferred under the BDS model. Errors were calculated per 1 Myr steps as the absolute difference between the speciation/extinction rate of the BDS model and the MBD model divided by the rates of the BDS model. MAPEs were computed as the median of the errors across replicates. We used the median rates instead of means to limit the influence of outliers and to make rates comparable to the median BDS rates of Neubauer *et al.* [34]. To compare the fit of individual windows for the 20 Myr versus the 100 Myr approach, median MAPEs across all 1 Myr steps were computed for the corresponding parts of the 100 Myr window. Effective sampling sizes and MAPEs were

calculated in the statistical programming environment R v. 4.0.3 [40] using the package 'coda' v. 0.19-3 [41].

## (c) Factor selection

We chose 10 predictors to test for their potential influence on speciation and extinction rates in each window, including one biotic (diversity) and nine abiotic factors (mean annual temperature, annual precipitation, continental area, mean elevation range, average basin size and number of basins, TRI, mean geographical distance among fossil occurrences and a novel TCI; table 1; electronic supplementary material, tables S1.1–3). Diversity was inferred directly from the birth–death model, including a correction for sampling heterogeneity through time and across lineages, and automatically included in the MBD model [4]. The data sources and methods used to quantify the abiotic factors are detailed below.

Temperature, precipitation and their change over time have been proposed as affecting the diversity of freshwater gastropods, both for current [42–45] and extinct faunas [33,42,46]. The same is true for elevation [46,47]. Area size and isolation have been shown to impact diversification in isolated ecosystems, such as islands and mountain tops [23,48,49]. We here consider the continental area as a measure for habitable space. A larger area is assumed to offer more habitats and may accommodate more species and speciation events. Similarly, we include mean geographical distance, calculated as the average of all distances among all localities of a 1 Myr time slice, as a measure for (bio)-geographic isolation (electronic supplementary material, table S1.3). As another proxy for biogeographic separation, we use the number of basins containing fossil species, hypothesizing that more endemic species might evolve in geographically separated water bodies. In turn, mean basin size was considered as a measure for extinction risk—large basins can sustain larger or more (and potentially connected) water bodies that may buffer environmental changes more efficiently than small, isolated water bodies [50,51]. Finally, we used the mean TRI across cells containing fossil occurrences per time slice as a proxy for topographic variation. The TRI is widely used to quantify the elevational variation among neighbouring cells in a digital elevation grid [52]. A high mean TRI among cells containing gastropod faunas is interpreted here as a high small-scale topographic isolation. Details on the quantification of the predictor variables and data sources are supplied in electronic supplementary material, text and tables S1.1–1.3. Since most abiotic factors are

calculated from locality data, we carried out a subsampling test and repeated the MBD analyses for a subset of the data to test for the impact of spatial heterogeneity on the results (see electronic supplementary material, text and figure S1.5).

## (d) A novel estimate for tectonic complexity

The last abiotic factor—the TCI—is newly developed herein (electronic supplementary material, table S1.2). Because of considerable tectonic activity Europe's outline has changed massively over the past 100 Myr [53,54]. We hypothesize that a more complex shape, with many islands and peninsulas as well as high variation in the topographic relief, might be mirrored in more complex biogeographic patterns with (at least partially) isolated evolutionary dynamics, which in turn can affect diversification rates.

The TCI builds on the shoreline development index for lakes [55], which describes the perimeter (P) deviation from a perfect circle with the same area (A) as the lake (electronic supplementary material, figure S1.2). We transferred this ratio to the shape of a continent and added the average continental TRI to account for variation in the z-axis as well (electronic supplementary material, figure S1.3):

$$TCI = \frac{P}{2\sqrt{A\pi}} \, TRI, \qquad (2.1)$$

The index has the same unit as the input TRI (herein in metres) and theoretically ranges from zero (no complexity) to infinity (maximum complexity). The theoretical TCI for a continent with the least geographical complexity (i.e. a circle) would be defined solely by the TRI. The TCI for a continent with little elevational variation is controlled by its outline.

A similar approach is available from Zaffos et al. [56]. Their continental fragmentation index is based on a ratio between the perimeter of all tectonic plates in a merged state (i.e. plate overlaps removed) and the total perimeter of all plates. This method does not account for the actual area of land versus sea, which is why we chose to develop a new index that is explicitly focusing on the continental area.

## 3. Results

The diversification rates of European freshwater gastropods vary considerably over time (figure 1a). Notable peaks in speciation rate occur in the Late Cretaceous and at the Cretaceous–Palaeogene (K–Pg) boundary, and several smaller peaks appear during the Neogene. The extinction rate shows parallel spikes during the K–Pg boundary and the Neogene with similar magnitudes (the rates and the K–Pg boundary event, in particular, were described in detail by Neubauer et al. [34] and will not be discussed further here).

The reconstructed rates resulting from the MBD analyses for the 20 Myr windows largely match the diversification rates from the birth–death model with shifts (BDS), showing the same peaks with similar magnitudes (figure 1b). An exception is the speciation rate peak at the K–Pg boundary, which is much higher in the reconstructed rates. Nonetheless, median MAPEs show a small deviation from the BDS model, across the entire time span (speciation: 0.194, extinction: 0.165) as well as the individual windows (table 2). The overall good congruence between the original and reconstructed rates suggests that the chosen set of predictors accurately describes variation in the rates. Importantly, the congruence is better than for the analyses based on a single time window of 100 Myr (MAPEs: speciation: 0.331, extinction:

**4**

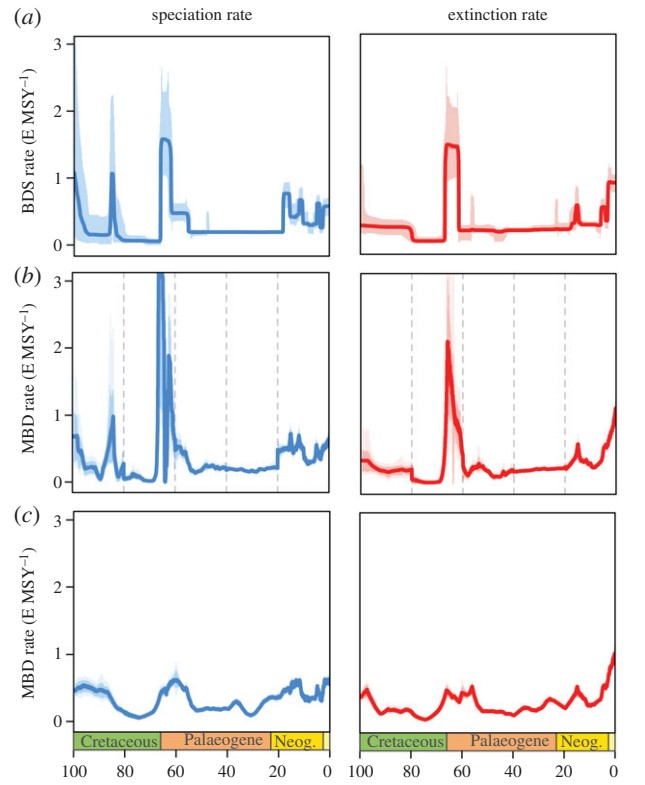

**Figure 1.** Diversification rates of European freshwater gastropods from (a) the original birth–death model with shifts (BDS; after Neubauer et al. [34]) and the reconstructed rates according to the multivariate birth–death models (MBD) for the (b) five 20 Myr windows and (c) the entire time frame of 100 Myr. Shaded areas indicate 95% credible intervals. E MSY$^{-1}$, events per million species years. (Online version in colour.)

**Table 2.** Deviation of the speciation and extinction rates reconstructed according to the multivariate birth–death models (MBD) from the original birth–death model with shifts (BDS; after Neubauer et al. [34]) for European freshwater gastropods. Deviation was quantified by the mean absolute percentage errors (MAPEs) for the total time frame of 100 Myr and individual time windows of 20 Myr duration, comparing the time-stratified approach (20 Myr) with time-invariant (100 Myr) effects of biotic and abiotic factors on rates.

| window | speciation rate | | extinction rate | |
|---|---|---|---|---|
| | 20 Myr approach | 100 Myr approach | 20 Myr approach | 100 Myr approach |
| total | 0.194 | 0.331 | 0.165 | 0.288 |
| 0–20 | 0.230 | 0.240 | 0.158 | 0.127 |
| 20–40 | 0.078 | 0.438 | 0.054 | 0.190 |
| 40–60 | 0.137 | 0.062 | 0.203 | 0.292 |
| 60–80 | 0.550 | 0.687 | 0.430 | 0.750 |
| 80–100 | 0.357 | 0.569 | 0.173 | 0.345 |

0.288; figure 1c). Despite following a similar trend, the peaks extend over longer time intervals and are of a much lesser magnitude in the 100 Myr approach. Slicing the rates obtained by the 100 Myr approach into windows of 20 Myr

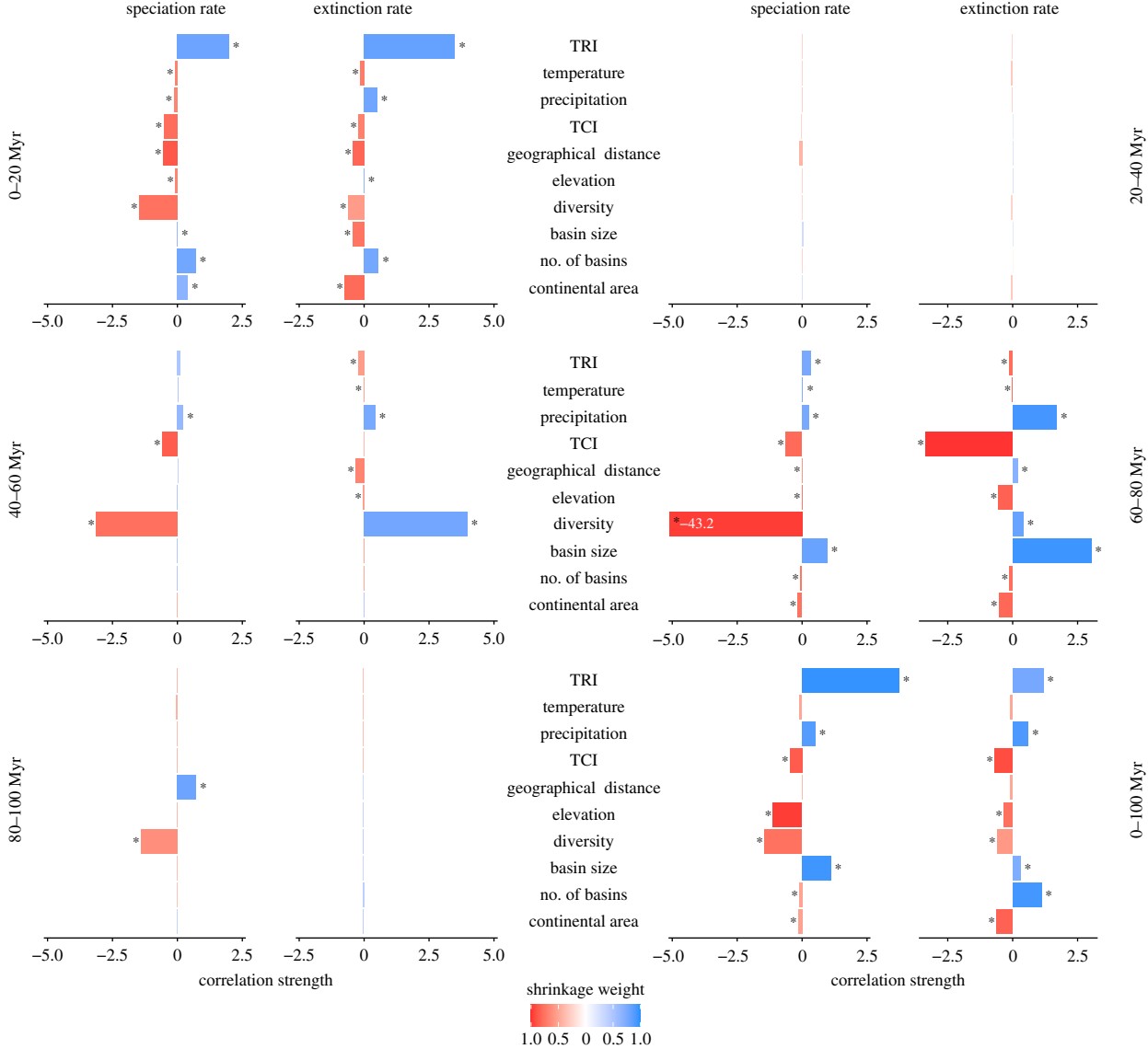

**Figure 2.** Influence of biotic and abiotic predictors on diversification dynamics of freshwater gastropods over different time windows. Significant relationships between predictors and rates (mean shrinkage weights greater than 0.5) are marked with asterisks. For display purpose, large values are truncated at |5| and shown on the respective bar. TRI, terrain ruggedness index; TCI, tectonic complexity index. (Online version in colour.)

duration returned better MAPEs in 2 out of 10 comparisons; in all other cases, the 20 Myr approach has a substantially better match with the rates inferred under the BDS model (table 2). This lower congruence supports our approach using multiple shorter time windows.

The combinations of predictors that significantly contribute to rate variation differ among the five windows (figure 2). For the 0–20 Myr and 60–80 Myr windows, all predictors show a significant influence (mean shrinkage weights greater than 0.5) on both speciation and extinction rates. However, only a few factors have a high effect size of influence and can thus be considered as relevant contributors to the model. TRI is the factor with the highest effect size for both speciation and extinction rates in the 0–20 Myr window. By contrast, diversity is by far the most important factor controlling speciation in the 60–80 Myr window, while a different set of factors (particularly the TCI, basin size and precipitation) have similarly high contributions to the extinction rate. No factor has a significant contribution in the 20–40 Myr window, which is expected given the constancy of the rates during this time frame compared to variation in the factors (electronic supplementary

material, tables S1.1–3). Diversity is the only important factor in the 40–60 Myr window, while the effect of other variables is minor (figure 2). For the 80–100 Myr window, only diversity and, to a lesser extent, geographical distance are relevant, but only for the speciation rate. All factors except temperature and geographical distance have significant impact on both speciation and extinction rate across the entire 100 Myr, whereas TRI has a strong influence on speciation.

## 4. Discussion

Our results show a strong variation in the effect of drivers of speciation and extinction rates through geological time. An interplay between biotic (diversity) and abiotic (particularly terrain ruggedness) factors influence speciation and extinction, yet their importance and direction of effect vary over the different time intervals for both rates (figure 2). There is no single factor that is relevant in all windows, although diversity-dependence shows a significant impact in most. The diversification rates obtained by dissecting the entire

time frame into five 20 Myr windows resemble the rates inferred with a flexible shift model better than those based on the entire 100 Myr time frame (figure 1 and table 2), supporting our hypothesis that the influence of drivers changes over time.

## (a) Drivers of diversification in European freshwater gastropods

Given the multitude of different factors and factor combinations that play a role in the diversification of European freshwater gastropods in different time windows, we will discuss in this section only the most important factors. Moreover, our study focuses mainly on the changing importance of drivers through time, rather than on the influence of individual factors.

The importance of diversity-dependence over most of the time and for both speciation and extinction rates is congruent with previous studies dealing with different animal and plant clades, time intervals and geographical regions [1,2,4,22,26,57–59]. These studies involve a number of invertebrate taxa, including marine gastropods [60]. Diversity-dependence is a macroevolutionary equivalent for species competition (i.e. higher species richness is expected to increase competition for habitat space, resources or niche space in general) [2,10–12]. In turn, competition is suggested to decrease the rate at which new species originate and increase the rate at which species go extinct. A recent example was demonstrated by Wilke *et al*. [13] for lacustrine diatoms, showing an asymptotic increase of species richness as a (partial) result of a diversity-dependent decrease in the speciation rate. In our case, the strong control of diversity over both extinction and speciation rate in various time periods suggests that species competition is an overarching factor shaping diversification in freshwater gastropods over long geological time scales, but with a differing effect sizes.

For the last 20 Myr and also for the entire time frame, topographic isolation (quantified here as TRI) plays a vital role for both speciation and extinction rates. The positive correlation indicates that a high topographic variability drives an increase in both rates. The last 20 Myr—largely corresponding to the Neogene and Quaternary periods—coincide with the development of numerous long-lived lakes in central and south-eastern Europe, which sparked the evolution of diverse and highly endemic faunas [33,61]. The geographical isolation paired with the geological long-term stability of these basins and lakes sets an ideal stage for intralacustrine (or intrabasinal) speciation, which probably led to the observed increase in the speciation rate linked to TRI. At the same time, geographically isolated faunas are typically more vulnerable to climatic or environmental changes [50,51]. Eventually, all of the Neogene faunas went extinct with the disappearance of the respective lakes. This high and geologically rapid turnover is reflected in the increased species turnover (i.e. a high ratio of extinction to speciation rate). Topographic isolation has also been shown as an important driver of diversification in other isolated ecosystems, such as mountains and islands [23,48,49].

Other factors play only minor roles through most of the time. Only for the 60–80 Myr window, additionally relevant factors seem to be basin size (for speciation and extinction rates) and our novel TCI (for extinction). However, the effect of these factors must be interpreted with caution. This window contains the K–Pg mass extinction event, which is the ultimate cause of most species extinctions in that interval [34,62]. Such a rapid, catastrophic event is unlikely to be captured accurately by the chosen set of predictors, and we therefore avoid any interpretation of single drivers.

## (b) Methodological implications and limitations

Our approach analysing the effect of abiotic and biotic factors on diversification rates in several shorter time windows has advantages over the standard procedure of using a single, long time frame. The diverse set of predictors that influences rates across the entire 100 Myr time frame is likely to be the result of a confounding effect of multiple processes acting at different times and over different time scales. For example, the strong positive influence of TRI on the speciation rate may be a result of the effect in the 0–20 Myr window (figure 2). Conversely, the strong negative impact of diversity-dependence found in several shorter windows is not well reflected in the results for the entire time frame. As such, this single time window approach struggles to disentangle the impact of different factors on diversification rates, which act over short and long time scales. This problem is also reflected by the poor match between the speciation and extinction rates inferred through a BDS model and the reconstructed rates from the MBD model for the entire time frame (figure 1 and table 2). These findings provide strong support for our time-varying approach.

Naturally, the set of factors and their influence on diversification rates might vary even within the shorter time windows of 20 Myr. For example, the apparent lack of a major influence of climate on long-term diversification in our analyses is likely a result of the chosen time frames (or, alternatively, the large geographical area, bridging several climate zones and ecoregions). Freshwater gastropod faunas are known to respond to climate change on much shorter time scales, in the order of tens to thousands of years [42,46]. Unfortunately, the stratigraphic uncertainties of the fossil record do not allow increasing the temporal resolution of the analyses, at least not over the entire time scale.

The optimal window size probably varies across species groups and depends especially on the stratigraphic resolution of the fossil record and the potential drivers. Moreover, the spatial scale plays a vital role. Whether single ecosystems or entire continents are assessed has an influence on the evolutionary relationships among the species included in the analyses. Optimally, future studies should further develop our approach and test for a link between rates and biotic and abiotic factors across windows of variable duration and starting points as well as geographical scales. This way, variation in the drivers and the spatio-temporal dimensions they are relevant for can be specified more precisely.

The current limitations of our approach involve the interdependence across factors (e.g. basin size change and topographic changes). Although collinearity among variables does not bias the MBD model, it complicates identifying the exact nature and degree of the relationships with the predictors (electronic supplementary material, figure S1.4). Additionally, unknown and therefore unaccounted for factors might also affect the speciation and extinction dynamics without being detected. Future improvements of the

method may attempt to incorporate potential confounding effects and interaction terms as well as explicit variable selection to alleviate these problems.

The novel TCI showed a significant relationship with both speciation and extinction rates and as such provides a relevant addition to our toolkit. Future studies need to test the relevance of this factor for the diversification history of other groups, particularly of terrestrial biota. Beyond its application as an evolutionary driver, it may prove a useful tool to quantify and compare geographical features. As for other large-scale topographic proxies, the general limitation of the TCI is linked to the resolution and accuracy of the palaeogeographic reconstructions. Despite our effort to account for uncertainties in the shape of the coastlines, features smaller than 50 km are not resolvable on a palaeogeographic map. The resolution of the topographic grid used herein produces a false sense of accuracy. Nonetheless, we are confident that factors inferred on global or continental scales, such as in our analyses, are unlikely to suffer from major biases. As such, this index can provide a useful approximation of general trends and variation in tectonic complexity.

Data accessibility. Data used in this study are available in the electronic supplementary material [63] and from the public repository of the JLU Giessen: http://doi.org/10.22029/jlupub-9.

Authors' contributions. T.A.N.: conceptualization, formal analysis, funding acquisition, investigation, methodology, project administration, software, validation, visualization, writing—original draft, writing—review and editing; T.H.: formal analysis, software, writing—review and editing; D.S.: formal analysis, software, writing—review and editing; C.R.S.: data curation, resources; B.S.: formal analysis, writing—review and editing; C.A.: investigation; D.D.: investigation, writing—review and editing; M.H.: investigation; T.W.: conceptualization, writing—review and editing.

All authors gave final approval for publication and agreed to be held accountable for the work performed therein.

Competing interests. The authors declare no competing interests.

Funding. T.A.N. was supported by the Deutsche Forschungsgemeinschaft (DFG, grant no. NE 2268/2-1), the Justus Liebig University Giessen (Just'us Fellowship) and an Alexander-von-Humboldt Fellowship; D.S. and T.H. were funded by the Swiss National Science Foundation (grant no. PCEFP3_187012); D.S. also acknowledges funding from the Swedish Research Council (grant no. VR: 2019-04739); B.S. and T.W. were supported by the DFG (grant nos. STE 2460/4-1 and WI 1902/17-1).

Acknowledgements. We are grateful to Burkhard Linke and Alexander Goesmann for assistance during the cloud setup; to Benjamin Mills for providing climate data; to Philipp Kraft for advice on using the modelling builder in ArcGIS; to Frank P. Wesselingh for lively discussions; and to Wolfgang Brunnbauer, Sebastian Calzada, Daniela Esu, Joachim Gründel, Sonja Herzog-Gutsch, Olaf Höltke, Arie W. Janssen, Ronald Janssen, Eike Neubert, Jean-Michel Pacaud, Sophie Passot, Simon Schneider, Maxim Vinarski and the colleagues at the Naturalis Biodiversity Center library for providing literature. Computational resources were provided by the BMBF-funded de.NBI Cloud within the German Network for Bioinformatics Infrastructure (de.NBI). We thank Peter J. Wagner, an anonymous reviewer and the associate editor for insightful feedback on this study.

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
