## [Peer Review File · Proceedings of the Royal Society B: Biological Sciences]

Review History

RSPB-2021-2057.R0 (Original submission)

Review form: Reviewer 1

Recommendation

Major revision is needed (please make suggestions in comments)

Scientific importance: Is the manuscript an original and important contribution to its field?

Good

General interest: Is the paper of sufficient general interest?

Excellent

Quality of the paper: Is the overall quality of the paper suitable?

Good

Is the length of the paper justified?

Yes

Should the paper be seen by a specialist statistical reviewer?

No

Do you have any concerns about statistical analyses in this paper? If so, please specify them explicitly in your report.

No

It is a condition of publication that authors make their supporting data, code and materials available - either as supplementary material or hosted in an external repository. Please rate, if applicable, the supporting data on the following criteria.

Is it accessible?

Yes

Is it clear?

Yes

Is it adequate?

Yes

Do you have any ethical concerns with this paper?

No

Comments to the Author

In this manuscript, Neubauer et al. used paleontological models of species diversification to investigate the drivers of diversification of European freshwater gastropods. By focusing on different time windows, they reported that the relative influence of different abiotic and biotic diversification drivers can change through time.

Overall, I think this manuscript is very interesting and timely. It's also well written, although the authors use many acronyms that complexify the reading. I hope my comments will be constructive and helpful.

I am not an expert on paleontological diversification models but I do have two concerns/questions about how the results could be impacted by (i) the incomplete and heterogenous sampling and (ii) the interdependence across factors.

First, I am wondering how incomplete and heterogenous sampling can bias the results. For instance, both the mean geographic distance the biogeographic separation are two variables that can be largely influenced by sampling bias. Does the model explicitly take that into account? If not, I would suggest some rarefaction/subsampling of the datasets to assess how the conclusions are robust to the heterogeneity and the incompleteness of the fossil records.

Second, the authors acknowledge that the "current limitations of our approach involve the interdependence across factors (e.g. basin size change and topographic changes)" (l. 366-367). I think they should at least show the correlations between the different variables tested, and if some of them are strongly correlated, they should rerun the models with only uncorrelated variables to assess how interdependence across factors can influence their results and their conclusions.

Other comments:

- It's unclear in the abstract if the authors performed additional developments of the multivariate birth-death model or not. In addition, they should explicitly write that they are using paleontological data only.

- l. 43-49: the authors should maybe also refer to species interactions other than “density dependence” (i.e. competition within a clade) and mention that trophic or mutualistic interactions can also drastically affect species diversification.

- l. 56-60: True, but it also exists some extensions of these models that allow testing for the effect of a given factor on diversification at a given slice of time only (see Lewitus et al., 2018).

- l. 60-61: “For example, geological processes like continental fragmentation likely vary at a slower pace than biotic interactions and could therefore mask their effects.” It’s unclear to me how this can “mask their effect”? Both would be confounded, but I don’t see how the effect of continental fragmentation can be unraveled in this case, even if looking at short time windows. This should be clarified.

- l. 78-89: I think that this paragraph about the “choice of the taxonomic level” and how to obtain “reasonable estimates of diversification rates” is quite unclear. For instance, why an ideal model taxon for providing reasonable estimates of diversification rates should have representative coverage of different habitat types and lifestyles? Actually, if the taxa are present in different habitat types and lifestyles, these habitat types and lifestyles should indeed have a representative coverage, but not the other way.

- Given that the Multivariate birth-death analyses are central to the paper, their principle could be better explained in the Methods section, and for instance, the details about the parameters of the MCMC runs could be put in the supplementary material.

- I feel they are too many acronyms in the Result section, figure legends, and in the discussion (BDS, MBD, MAPEs...). They should be avoided or at least be redefined every time.

- l. 267: “No parameter has a significant contribution in the 20–40 Myr window, which is expected given the homogeneity of the rates during this time frame”: I don’t see why no factors are expected to have a contribution. Indeed, if the factor driving the species diversification is also constant, it can explain why rates are constant.

- l. 340-342: This sentence is quite unclear to me.

References:

Lewitus E, Bittner L, Malviya S, Bowler C, Morlon H. 2018. Clade-specific diversification dynamics of marine diatoms since the Jurassic. *Nature Ecology and Evolution* 2: 1715–1723.

Review form: Reviewer 2

Recommendation

Accept with minor revision (please list in comments)

Scientific importance: Is the manuscript an original and important contribution to its field?

Excellent

General interest: Is the paper of sufficient general interest?

Good

Quality of the paper: Is the overall quality of the paper suitable?

Excellent

Is the length of the paper justified?

Yes

Should the paper be seen by a specialist statistical reviewer?

No

Do you have any concerns about statistical analyses in this paper? If so, please specify them explicitly in your report.

No

It is a condition of publication that authors make their supporting data, code and materials available - either as supplementary material or hosted in an external repository. Please rate, if applicable, the supporting data on the following criteria.

Is it accessible?

Yes

Is it clear?

Yes

Is it adequate?

Yes

Do you have any ethical concerns with this paper?

No

Comments to the Author

Although there are numerous analyses of drivers of diversification for marine invertebrates, such studies are still remarkably uncommon for terrestrial invertebrate groups with the possible exception of insects. Neubauer et al. attempt to partially rectify this by analyzing diversification dynamics of European freshwater gastropods over the last 100 million years. They do this by applying Bayesian birth-death-sampling analyses (PyRate) to a database of nearly 25K occurrences of over 3K species from nearly 6K localities. Using three related approaches, the authors find very high extinction rates at the end of the Cretaceous followed by elevated origination rates in the early Cenozoic, fairly low diversification through the remainder of the Paleogene, and then elevated volatility in the Neogene with both origination and (to a lesser extent) extinction rates showing multiple pulses.

Given the long history of using marine invertebrates and terrestrial vertebrates for these sorts of analyses, this is a very welcome addition to diversification studies. The analyses employ very modern and sophisticated tools that treats constant rates through time as a Bayesian analog of a null hypothesis, including of a multivariate analysis that allows assessment of the impact of particular extrinsic parameters on the diversification dynamics of freshwater snails. This stands out from what we usually see in studies of drivers of diversification in that the association is directly modeled rather than used as some inflection point (e.g., rates before and after the Paleocene-Eocene Thermal Maximum).

Overall, I think that this is an excellent study. My sole concerns are that the results and discussion occur largely in a vacuum relative to comparable studies. Something that I think is a real selling point of this study is that it is using gastropods: and we have a lot of diversity studies for marine gastropods for the entire Phanerozoic. Quite a few studies examine Meso-Cenozoic gastropod dynamics: for example, I think that studies using data from the Paleobiology Database (e.g., Alroy 2010 *Science* 329:1191) find diversification dynamics among marine gastropods not dissimilar to those shown here. Regardless, gastropods are one of the few groups for which we can do this with both marine and terrestrial fossils, and that makes for interesting contrasts of evolutionary dynamics. Similarly, it might be worth contrasting these findings with those for

mammals. We have a lot of studies looking at mammal diversification, including those from the same region over the same timespan (e.g., Finarelli & Liow 2016 *Biol. J. Linn. Soc.* 118:26). In particular, it jumps out to me that volatility among freshwater snails increases during the Neogene: i.e., the time in which grasslands took over much of the planet and radically altered terrestrial ecosystems. We usually focus on what this did to “truly” terrestrial animals; however, grasslands must have affected the ecologies of lacustrine environments, too.

However, these are fairly small points that I would like to see added at the outset and in the discussion: neither merits more than a paragraph.

I have the additional particular comments.

Lines 62 – 65: “Little is understood about the changing impact of diversification drivers over time, and the available knowledge is based on incompatible methodological approaches (fossils vs. phylogenies, single clades vs. entire faunas, empirical data vs. simulations), geographic scales (single ecosystem vs. global) and species groups.”

I take issue with this statement! For example, a quick Google Scholar search about the relationship between environmental changes and shifting diversification dynamics for the Ordovician Radiations (i.e., “the Great Ordovician Biodiversification Event”) revealed several papers from 2021 alone. I chose that because it is near and dear to my research interests: but study of the potential drivers of marine diversification from throughout the Phanerozoic (and even Ediacaran) really are quite common. Now, what are not common are comparable studies for terrestrial invertebrates, particularly terrestrial invertebrates that are not insects.

Lines 115-117: We applied the Reversible Jump Markov Chain Monte Carlo (RJ-MCMC) algorithm using the birth–death model with shifts (BDS), i.e. speciation and extinction rates are allowed to shift across time frames [35].

I have two comments/questions here. First, PyRate should allow sampling rates to vary through time, too. Assuming that is done, then it should be mentioned. Second, it might be worth noting what analyses are (Reversibly) Jumping between. For standard PyRate analyses, that usually would be an N-parameter model to an N-1 or N+1 parameter model (e.g., one more or fewer shifts in diversification or sampling). With the multivariate analysis, I would guess that it includes jumps from N associations between diversification and an extrinsic parameter to N+1 or N-1 such associations. Regardless, both should be stated. (RJMCMC is becoming widespread: but I don’t know that we are to the point where we can assume that readers know what it does in the way that we can probably assume that they know what a principal components or parsimony analysis does.)

Lines 163-164: “We chose ten parameters to test for their potential influence on speciation and extinction rates in each window, including one biotic (diversity) and nine abiotic factors...”

I almost hate to open this can of worms, but did the authors consider biotic parameters such as the diversification of mammal groups or fish groups, or the diversification of grasses? Given that they find diversity dependence (i.e., “logistic” in the paleo literature!) and also an effect of topography, this might not be too important; however, one question that jumped to my mind was whether the diversification of any other taxa commonly found in freshwater habitats or directly affecting them (as grasslands certainly must have done) coincide with changes in freshwater gastropod diversification. One might not be surprised to see increased volatility given some “Red Queen” scenario in which new additions to the ecoscape generated arms races with the snails in one capacity or another.

Lines 289 – 291: “There is no single parameter that is relevant in all windows, although diversity-dependence shows a significant impact in most.”

That's cool! It is also, I think, worth noting that diversity-dependence (or, really, richness-dependence) is found in a lot of marine invertebrate studies, including some including gastropods.

These are, I think, fairly minor quibbles/comments/questions. The final one in particular might well be the topic of a completely different paper.

Sincerely,
Peter J. Wagner
University of Nebraska, Lincoln

Alroy, J. 2010. The shifting balance of diversity among major marine animal groups. *Science* 329:1191–1194. (10.1126/science.1189910)

Finarelli, J. A. and L. H. Liow. 2016. Diversification histories for North American and Eurasian carnivorans. *Biol. J. Linn. Soc.* 118:26 – 38. (10.1111/bij.12777)

(I have attached a pdf version which is easier to read!)

Decision letter (RSPB-2021-2057.R0)

01-Nov-2021

Dear Dr Neubauer:

Your manuscript has now been peer reviewed and the reviews have been assessed by an Associate Editor. The reviewers' comments (not including confidential comments to the Editor) are included at the end of this email for your reference. As you will see, the reviewers have raised some concerns with your manuscript and we would like to invite you to revise your manuscript to address them. One of the referees suggested that the broader significance of your work would be better appreciated if you are able to show that although diversification studies of marine invertebrates (including marine gastropods) have been quite common, diversification studies of terrestrial invertebrates other than insects have not. This is in contrast to terrestrial gastropods which are a pretty successful group, and they play pretty prominent roles in a lot of terrestrial ecosystems. There is an opportunity here to make a comparison between terrestrial and marine systems.

Research ethics:

Use of animals and field studies:

It is a condition of publication that you make available the data and research materials supporting the results in the article. Please see our Data Sharing Policies (<https://royalsociety.org/journals/authors/author-guidelines/#data>). Datasets should be deposited in an appropriate publicly available repository and details of the associated accession number, link or DOI to the datasets must be included in the Data Accessibility section of the article (<https://royalsociety.org/journals/ethics-policies/data-sharing-mining/>). Reference(s) to datasets should also be included in the reference list of the article with DOIs (where available).

Please submit a copy of your revised paper within three weeks. If we do not hear from you within this time your manuscript will be rejected. If you are unable to meet this deadline please let us know as soon as possible, as we may be able to grant a short extension.

Best wishes,
Dr Daniel Costa
mailto: proceedingsb@royalsociety.org

Associate Editor Comments:

Neubauer and colleagues use a multivariate birth-death model to estimate drivers of diversification for European freshwater gastropod species over the past 100 million years. They find that the parameters affecting origination and extinction rates vary considerably over this time. The authors' analyses appear robust, and both reviewers were positive about the novelty and appeal of this contribution for Proc B. Although complimentary, both reviewers also provided suggestions to help improve the clarity of this contribution. In addition to the excellent suggestions by both reviewers, I provide additional comments below.

Line 23: suggest changing to: 'for our understanding'

Line 25: this sentence is a bit confusing. I understand what you mean, but I think some rewriting would help. Essentially, you mean that studies have not examined how abiotic and biotic drivers may change over an interval of time.

Line 31: remove 'a'

Line 42: have been (versus has)

Line 44: consider rewording to 'abiotic factors proposed to affect diversification' (or something similar). You need a modifier here.

Line 50: 'the many potential controls' is simpler and easier to read

Line 59-61: I am not sure I understand what you mean here. I would argue that a longer-term study would pick up the effects of continental fragmentation because this process is slow. To me, it seems that shorter term fluctuations would be flagged more readily with shorter temporal windows. Could you please clarify?

Line 74: controlled – past tense.

Line 76: suggest rewording to 'we investigate potential variation in the influence of abiotic and biotic drivers'

Line 79: have been performed primarily

Line 97-102: this is a very long sentence, and I would suggest rewording or breaking into two for clarity.

Line 107: we also evaluate how drivers differ across time windows to assess...

Line 117-119: The authors refer to a previous contribution for details of data selection, geographic scope, and analysis. However, I think this needs to be recapitulated in the present manuscript. The details can be provided in a supplement. That said, the authors should more explicitly list the

abiotic and biotic variables in the main Methods. At present, it is difficult to understand precisely what abiotic and biotic predictors were used, and how these predictors were estimated. Providing these in list form or in a table in the main manuscript may help.

Regarding the predictors, how was diversity estimated? Did the authors use SQS, and did they account for sampling biases? I could not find this information in the manuscript or supplement. Was diversity estimated across the entire study area, or constrained to the regions in which speciation and extinction were occurring in the particular windows?

I also wondered about how other variables were estimated: were they taken from a particular region where speciation/extinction occurred within a given time window, or were they always estimated across all of the study region for all time windows (or across the occurrences for a time window)?

Line 128: specify here what type of rates you mean (both extinction/speciation, presumably)

Line 140-141: I wondered how the authors came to choose the 20 myr temporal windows. They indicate that they ran test analyses, but what were the criteria for choosing a window for these analyses? And, did the authors also consider other temporal windows? How might results differ if they restrict or expand this temporal resolution (aside from the 20 myr and 100 myr windows already run)?

Line 145: For analyses with low ESS, did the authors remove these from analysis? This was not stated.

Line 234: I would not be surprised if there was a peak in turnover at the K-Pg boundary, but I worry this effect might result from missing time / rock record. Can the authors confirm or speak to whether this peak is real or simply reflects the architecture of the stratigraphic record?

Line 267: the authors should explain more why no parameter is expected to have a significant contribution in the 20-40 myr bin.

Line 290: for both speciation and extinction? Or just origination? It would be useful to specify here.

Reviewer(s)' Comments to Author:

Referee: 1

Comments to the Author(s)

In this manuscript, Neubauer et al. used paleontological models of species diversification to investigate the drivers of diversification of European freshwater gastropods. By focusing on different time windows, they reported that the relative influence of different abiotic and biotic diversification drivers can change through time.

Overall, I think this manuscript is very interesting and timely. It's also well written, although the authors use many acronyms that complexify the reading. I hope my comments will be constructive and helpful.

I am not an expert on paleontological diversification models but I do have two concerns/questions about how the results could be impacted by (i) the incomplete and heterogenous sampling and (ii) the interdependence across factors.

First, I am wondering how incomplete and heterogenous sampling can bias the results. For instance, both the mean geographic distance the biogeographic separation are two variables that can be largely influenced by sampling bias. Does the model explicitly take that into account? If

not, I would suggest some rarefaction/subsampling of the datasets to assess how the conclusions are robust to the heterogeneity and the incompleteness of the fossil records.

Second, the authors acknowledge that the “current limitations of our approach involve the interdependence across factors (e.g. basin size change and topographic changes)” (l. 366-367). I think they should at least show the correlations between the different variables tested, and if some of them are strongly correlated, they should rerun the models with only uncorrelated variables to assess how interdependence across factors can influence their results and their conclusions.

Other comments:

- It's unclear in the abstract if the authors performed additional developments of the multivariate birth-death model or not. In addition, they should explicitly write that they are using paleontological data only.

- l. 43-49: the authors should maybe also refer to species interactions other than “density dependence” (i.e. competition within a clade) and mention that trophic or mutualistic interactions can also drastically affect species diversification.

- l.56-60: True, but it also exists some extensions of these models that allow testing for the effect of a given factor on diversification at a given slice of time only (see Lewitus et al., 2018).

- l. 60-61: “For example, geological processes like continental fragmentation likely vary at a slower pace than biotic interactions and could therefore mask their effects.” It's unclear to me how this can “mask their effect”? Both would be confounded, but I don't see how the effect of continental fragmentation can be unraveled in this case, even if looking at short time windows. This should be clarified.

- l. 78-89: I think that this paragraph about the “choice of the taxonomic level” and how to obtain “reasonable estimates of diversification rates” is quite unclear. For instance, why an ideal model taxon for providing reasonable estimates of diversification rates should have representative coverage of different habitat types and lifestyles? Actually, if the taxa are present in different habitat types and lifestyles, these habitat types and lifestyles should indeed have a representative coverage, but not the other way.

- Given that the Multivariate birth-death analyses are central to the paper, their principle could be better explained in the Methods section, and for instance, the details about the parameters of the MCMC runs could be put in the supplementary material.

- I feel they are too many acronyms in the Result section, figure legends, and in the discussion (BDS, MBD, MAPEs...). They should be avoided or at least be redefined every time.

- l. 267: “No parameter has a significant contribution in the 20–40 Myr window, which is expected given the homogeneity of the rates during this time frame”: I don't see why no factors are expected to have a contribution. Indeed, if the factor driving the species diversification is also constant, it can explain why rates are constant.

- l. 340-342: This sentence is quite unclear to me.

References:

Lewitus E, Bittner L, Malviya S, Bowler C, Morlon H. 2018. Clade-specific diversification dynamics of marine diatoms since the Jurassic. *Nature Ecology and Evolution* 2: 1715–1723.

Referee: 2

Comments to the Author(s)

Although there are numerous analyses of drivers of diversification for marine invertebrates, such studies are still remarkably uncommon for terrestrial invertebrate groups with the possible exception of insects. Neubauer et al. attempt to partially rectify this by analyzing diversification dynamics of European freshwater gastropods over the last 100 million years. They do this by applying Bayesian birth-death-sampling analyses (PyRate) to a database of nearly 25K occurrences of over 3K species from nearly 6K localities. Using three related approaches, the authors find very high extinction rates at the end of the Cretaceous followed by elevated origination rates in the early Cenozoic, fairly low diversification through the remainder of the Paleogene, and then elevated volatility in the Neogene with both origination and (to a lesser extent) extinction rates showing multiple pulses.

Given the long history of using marine invertebrates and terrestrial vertebrates for these sorts of analyses, this is a very welcome addition to diversification studies. The analyses employ very modern and sophisticated tools that treat constant rates through time as a Bayesian analog of a null hypothesis, including of a multivariate analysis that allows assessment of the impact of particular extrinsic parameters on the diversification dynamics of freshwater snails. This stands out from what we usually see in studies of drivers of diversification in that the association is directly modeled rather than used as some inflection point (e.g., rates before and after the Paleocene-Eocene Thermal Maximum).

Overall, I think that this is an excellent study. My sole concerns are that the results and discussion occur largely in a vacuum relative to comparable studies. Something that I think is a real selling point of this study is that it is using gastropods: and we have a lot of diversity studies for marine gastropods for the entire Phanerozoic. Quite a few studies examine Meso-Cenozoic gastropod dynamics: for example, I think that studies using data from the Paleobiology Database (e.g., Alroy 2010 *Science* 329:1191) find diversification dynamics among marine gastropods not dissimilar to those shown here. Regardless, gastropods are one of the few groups for which we can do this with both marine and terrestrial fossils, and that makes for interesting contrasts of evolutionary dynamics. Similarly, it might be worth contrasting these findings with those for mammals. We have a lot of studies looking at mammal diversification, including those from the same region over the same timespan (e.g., Finarelli & Liow 2016 *Biol. J. Linn. Soc.* 118:26). In particular, it jumps out to me that volatility among freshwater snails increases during the Neogene: i.e., the time in which grasslands took over much of the planet and radically altered terrestrial ecosystems. We usually focus on what this did to “truly” terrestrial animals; however, grasslands must have affected the ecologies of lacustrine environments, too.

However, these are fairly small points that I would like to see added at the outset and in the discussion: neither merits more than a paragraph.

I have the additional particular comments.

Lines 62 – 65: “Little is understood about the changing impact of diversification drivers over time, and the available knowledge is based on incompatible methodological approaches (fossils vs. phylogenies, single clades vs. entire faunas, empirical data vs. simulations), geographic scales (single ecosystem vs. global) and species groups.”

I take issue with this statement! For example, a quick Google Scholar search about the relationship between environmental changes and shifting diversification dynamics for the Ordovician Radiations (i.e., “the Great Ordovician Biodiversification Event”) revealed several papers from 2021 alone. I chose that because it is near and dear to my research interests: but study of the potential drivers of marine diversification from throughout the Phanerozoic (and even Ediacaran) really are quite common. Now, what are not common are comparable studies for terrestrial invertebrates, particularly terrestrial invertebrates that are not insects.

Lines 115-117: We applied the Reversible Jump Markov Chain Monte Carlo (RJ-MCMC) algorithm using the birth–death model with shifts (BDS), i.e. speciation and extinction rates are allowed to shift across time frames [35].

I have two comments/questions here. First, PyRate should allow sampling rates to vary through time, too. Assuming that is done, then it should be mentioned. Second, it might be worth noting what analyses are (Reversibly) Jumping between. For standard PyRate analyses, that usually would be an N-parameter model to an N-1 or N+1 parameter model (e.g., one more or fewer shifts in diversification or sampling). With the multivariate analysis, I would guess that it includes jumps from N associations between diversification and an extrinsic parameter to N+1 or N-1 such associations. Regardless, both should be stated. (RJMCMC is becoming widespread: but I don't know that we are to the point where we can assume that readers know what it does in the way that we can probably assume that they know what a principal components or parsimony analysis does.)

Lines 163-164: “We chose ten parameters to test for their potential influence on speciation and extinction rates in each window, including one biotic (diversity) and nine abiotic factors...”

I almost hate to open this can of worms, but did the authors consider biotic parameters such as the diversification of mammal groups or fish groups, or the diversification of grasses? Given that they find diversity dependence (i.e., “logistic” in the paleo literature!) and also an effect of topography, this might not be too important; however, one question that jumped to my mind was whether the diversification of any other taxa commonly found in freshwater habitats or directly affecting them (as grasslands certainly must have done) coincide with changes in freshwater gastropod diversification. One might not be surprised to see increased volatility given some “Red Queen” scenario in which new additions to the ecoscape generated arms races with the snails in one capacity or another.

Lines 289 – 291: “There is no single parameter that is relevant in all windows, although diversity-dependence shows a significant impact in most.”

That's cool! It is also, I think, worth noting that diversity-dependence (or, really, richness-dependence) is found in a lot of marine invertebrate studies, including some including gastropods.

These are, I think, fairly minor quibbles/comments/questions. The final one in particular might well be the topic of a completely different paper.

Sincerely,
Peter J. Wagner
University of Nebraska, Lincoln

Alroy, J. 2010. The shifting balance of diversity among major marine animal groups. *Science* 329:1191–1194. (10.1126/science.1189910)

Finarelli, J. A. and L. H. Liow. 2016. Diversification histories for North American and Eurasian carnivorans. *Biol. J. Linn. Soc.* 118:26 – 38. (10.1111/bij.12777)

(I have attached a pdf version which is easier to read!)

Author's Response to Decision Letter for (RSPB-2021-2057.R0)

See Appendix A.

RSPB-2021-2057.R1 (Revision)

Review form: Reviewer 1

Recommendation

Accept as is

Scientific importance: Is the manuscript an original and important contribution to its field?

Good

General interest: Is the paper of sufficient general interest?

Good

Quality of the paper: Is the overall quality of the paper suitable?

Good

Is the length of the paper justified?

Yes

Should the paper be seen by a specialist statistical reviewer?

No

Do you have any concerns about statistical analyses in this paper? If so, please specify them explicitly in your report.

No

It is a condition of publication that authors make their supporting data, code and materials available - either as supplementary material or hosted in an external repository. Please rate, if applicable, the supporting data on the following criteria.

Is it accessible?

Yes

Is it clear?

Yes

Is it adequate?

Yes

Do you have any ethical concerns with this paper?

No

Comments to the Author

The authors have answered all my questions and suggestions and the manuscript appears very good to my eyes.

Just a very minor typo: the references 8 and 28 are the same.

Decision letter (RSPB-2021-2057.R1)

06-Jan-2022

Dear Dr Neubauer

I am pleased to inform you that your Review manuscript RSPB-2021-2057.R1 entitled "Drivers of diversification in freshwater gastropods vary over deep time" has been accepted for publication in Proceedings B.

The referee(s) do not recommend any further changes. Therefore, please proof-read your manuscript carefully and upload your final files for publication. Because the schedule for publication is very tight, it is a condition of publication that you submit the revised version of your manuscript within 7 days. If you do not think you will be able to meet this date please let me know immediately.

To upload your manuscript, log into <http://mc.manuscriptcentral.com/prsb> and enter your Author Centre, where you will find your manuscript title listed under "Manuscripts with Decisions." Under "Actions," click on "Create a Revision." Your manuscript number has been appended to denote a revision.

You will be unable to make your revisions on the originally submitted version of the manuscript. Instead, upload a new version through your Author Centre.

1) A text file of the manuscript (doc, txt, rtf or tex), including the references, tables (including captions) and figure captions. Please remove any tracked changes from the text before submission. PDF files are not an accepted format for the "Main Document".

2) A separate electronic file of each figure (tiff, EPS or print-quality PDF preferred). The format should be produced directly from original creation package, or original software format. Please note that PowerPoint files are not accepted.

3) Electronic supplementary material: this should be contained in a separate file from the main text and the file name should contain the author's name and journal name, e.g. `authorname_procb_ESM_figures.pdf`

All supplementary materials accompanying an accepted article will be treated as in their final form. They will be published alongside the paper on the journal website and posted on the online figshare repository. Files on figshare will be made available approximately one week before the accompanying article so that the supplementary material can be attributed a unique DOI. Please see: <https://royalsociety.org/journals/authors/author-guidelines/>

4) Data-Sharing and data citation

It is a condition of publication that data supporting your paper are made available. Data should be made available either in the electronic supplementary material or through an appropriate repository. Details of how to access data should be included in your paper. Please see <https://royalsociety.org/journals/ethics-policies/data-sharing-mining/> for more details.

<http://datadryad.org/submit?journalID=RSPB&manu=RSPB-2021-2057.R1> which will take you to your unique entry in the Dryad repository.

Once again, thank you for submitting your manuscript to Proceedings B and I look forward to receiving your final version. If you have any questions at all, please do not hesitate to get in touch.

Sincerely,
Dr Daniel Costa
Editor, Proceedings B
mailto:proceedingsb@royalsociety.org

Associate Editor
Comments to Author:

The authors have done a nice job with the revisions and have addressed most reviewer concerns. Reviewer 1 re-reviewed the contribution and is happy with the changes, as am I. I have only a few minor editorial comments below.

Line 42: consider changing to 'A major question in evolutionary biology is the processes that drive diversification'

Line 52: consider changing 'to' to 'on'

Line 78: remove 'the' before 'lacustrine diatoms'

Line 131: remove the 'a'

Line 142: this is a bit confusingly worded, please consider revising

Line 155: consider rewording to "to avoid biasing the results,"

Line 234: consider removing 'the' before 'speciation rate'

Line 270: I am not sure what is meant by 'especially' here

Line 296: starting this sentence with 'this also concerns' is a bit confusing to me, as I am not sure what is meant by 'this'

Line 305: remove 'the'

Line 306: place a comma before 'but with different...'

Reviewer(s)' Comments to Author:

Referee: 1

Comments to the Author(s)

The authors have answered all my questions and suggestions and the manuscript appears very good to my eyes.

Just a very minor typo: the references 8 and 28 are the same.

Author's Response to Decision Letter for (RSPB-2021-2057.R1)

See Appendix B.

Decision letter (RSPB-2021-2057.R2)

10-Jan-2022

Dear Dr Neubauer

I am pleased to inform you that your manuscript entitled "Drivers of diversification in freshwater gastropods vary over deep time" has been accepted for publication in Proceedings B.

Your article has been estimated as being 9 pages long. Our Production Office will be able to confirm the exact length at proof stage.

Data Accessibility section

Open Access

Paper charges

Sincerely,

Proceedings B

Appendix A

Reply to reviewers

Dear editor,

Thank you for the positive feedback and the chance to revise our manuscript. We followed the suggestions of the associate editor and the two reviewers carefully. Most of the points were adopted as requested, only in few instances we did not implement the suggestions. Below we provide a point-by-point reply (in blue) along with detailed explanation in case we do not follow a suggestion.

Kind regards,

Thomas A. Neubauer

Editor Comments:

Your manuscript has now been peer reviewed and the reviews have been assessed by an Associate Editor. The reviewers' comments (not including confidential comments to the Editor) are included at the end of this email for your reference. As you will see, the reviewers have raised some concerns with your manuscript and we would like to invite you to revise your manuscript to address them. One of the referees suggested that the broader significance of your work would be better appreciated if you are able to show that although diversification studies of marine invertebrates (including marine gastropods) have been quite common, diversification studies of terrestrial invertebrates other than insects have not. This is in contrast to terrestrial gastropods which are a pretty successful group, and they play pretty prominent roles in a lot of terrestrial ecosystems. There is an opportunity here to make a comparison between terrestrial and marine systems.

We are grateful for the positive feedback and appreciate the ideas very much. The suggested comparison with marine invertebrates (including marine gastropods) is very problematic though. The studies and diversification dynamics referred to by Referee 2 are concerned with absolute diversity and how it changes through time. Our study is, in contrast, about the dynamics and drivers of diversification processes over time, i.e. which biotic/abiotic factors control changes in diversification rates and how their impact varies through time. These aspects have not been studied in marine or terrestrial invertebrates, which does not allow for a reasonable comparison with our freshwater taxa. Please also see the comment on the suggestion of Referee 2 for more details.

Associate Editor Comments:

Neubauer and colleagues use a multivariate birth-death model to estimate drivers of diversification for European freshwater gastropod species over the past 100 million years. They find that the parameters affecting origination and extinction rates vary considerably over this time. The authors' analyses appear robust, and both reviewers were positive about the novelty and appeal of this contribution for Proc B. Although complimentary, both reviewers also provided suggestions to help improve the clarity of this contribution. In addition to the excellent suggestions by both reviewers, I provide additional comments below.

Line 23: suggest changing to: 'for our understanding'

Changed as requested.

Line 25: this sentence is a bit confusing. I understand what you mean, but I think some rewriting would help. Essentially, you mean that studies have not examined how abiotic and biotic drivers may change over an interval of time.

We reworded the sentence.

Line 31: remove 'a'

Changed as requested.

Line 42: have been (versus has)

Changed as requested.

Line 44: consider rewording to 'abiotic factors proposed to affect diversification' (or something similar). You need a modifier here.

Changed as requested.

Line 50: 'the many potential controls' is simpler and easier to read

Changed as requested.

Line 59-61: I am not sure I understand what you mean here. I would argue that a longer-term study would pick up the effects of continental fragmentation because this process is slow. To me, it seems that shorter term fluctuations would be flagged more readily with shorter temporal windows. Could you please clarify?

We thank the editor for the comment. This was obviously not clearly formulated, but it is indeed what we meant. We added “in studies across an extended geological interval” at the end of the sentence for clarity.

Line 74: controlled – past tense.

Changed as requested.

Line 76: suggest rewording to ‘we investigate potential variation in the influence of abiotic and biotic drivers’

Changed as requested.

Line 79: have been performed primarily

Changed as requested.

Line 97-102: this is a very long sentence, and I would suggest rewording or breaking into two for clarity.

Good point! We moved the insert concerning the details of the factors to a second sentence.

Line 107: we also evaluate how drivers differ across time windows to assess...

We changed the sentence to “We also evaluate how the influence of biotic and abiotic factors varies across time windows to assess specifically the relative roles of abiotic and biotic factors for the diversification of freshwater gastropods”.

Line 117-119: The authors refer to a previous contribution for details of data selection, geographic scope, and analysis. However, I think this needs to be recapitulated in the present manuscript. The details can be provided in a supplement. That said, the authors should more explicitly list the abiotic and biotic variables in the main Methods. At present, it is difficult to understand precisely what abiotic and biotic predictors were used, and how these predictors were estimated. Providing these in list form or in a table in the main manuscript may help.

We added the requested information on previous contributions and data in the electronic supplementary material. Also, we added a table in the main text that summarizes the predictors used for the analyses.

Regarding the predictors, how was diversity estimated? Did the authors use SQS, and did they account for sampling biases? I could not find this information in the manuscript or supplement. Was diversity estimated across the entire study area, or constrained to the regions in which speciation and extinction were occurring in the particular windows?

Diversity is estimated as a part of the birth–death (BDS) analyses and automatically included in the multivariate birth–death (MBD) analyses. This step already involves a correction of the quantified

diversity to account for sampling and preservation biases. We added this information to the Methods chapter (lines 175–177¹).

I also wondered about how other variables were estimated: were they taken from a particular region where speciation/extinction occurred within a given time window, or were they always estimated across all of the study region for all time windows (or across the occurrences for a time window)?

Predictor variables were quantified in different ways. Area and the novel Tectonic Complexity Index were inferred for entire Europe, the other factors were quantified based on the data obtained at fossil localities for a time window (e.g. mean of all temperature values at localities with fossil species). This avoids a bias towards data-poor regions (e.g., Northern Europe has hardly any fossil occurrences, including it in the inference of the climatic or topographic factors would certainly bias the results).

We provide this information in lines 191–194, 199–202 and 217–222, the electronic supplementary material, and added the information also in the new table 1 for clarification.

Since the revision turned out to be slightly too long (as assessed with the page estimator), we moved the details about how the parameters were assessed to the electronic supplementary material. The same was done for Figure 1, which is not essentially needed in the main text.

Line 128: specify here what type of rates you mean (both extinction/speciation, presumably)

Changed as requested.

Line 140-141: I wondered how the authors came to choose the 20 myr temporal windows. They indicate that they ran test analyses, but what were the criteria for choosing a window for these analyses? And, did the authors also consider other temporal windows? How might results differ if they restrict or expand this temporal resolution (aside from the 20 myr and 100 myr windows already run)?

Getting a reasonable time window length was the most difficult task of this study. We ran test analyses starting with a window length of 10 Myr, but those rarely converged even when using very high settings for the analyses. We added a sentence in the Methods to clarify this (lines 143–145).

Of course, there is no upper limit to the window length. We took the smallest achievable window size (i.e. 20 Myr) to assess our hypothesis (that biotic and abiotic drivers vary over time), but we actually plan in future projects to assess in greater detail how drivers change across time, and which are the time lengths relevant for which type of driver. However, these questions go beyond the scope of the present study. Considering the high number of windows, factors and replicates (accounting for the uncertainty in speciation and extinction times) needed to assess these questions, the computational effort to yield results based on sufficient sampling would be in the order of several months, even on a computational cloud.

Line 145: For analyses with low ESS, did the authors remove these from analysis? This was not stated.

¹ Note that all line numbers refer to the clean version of the revision.

Thanks for the comment! We re-calculated the correlation strengths and the MAPEs and revised the plots without the runs that yielded low ESS (which concerns only 26 out of 525 replicates in total). The values changed only slightly in a few cases, but the relative importance of the factors and, most importantly, the conclusions related to the correlation strengths and MAPEs were not affected. We replaced Figs. 2 and 3 and Table 1 (now Table 2) with the revised versions and rephrased the respective section in the Methods chapter.

Line 234: I would not be surprised if there was a peak in turnover at the K-Pg boundary, but I worry this effect might result from missing time / rock record. Can the authors confirm or speak to whether this peak is real or simply reflects the architecture of the stratigraphic record?

The fossil record of European freshwater gastropods contains data for several latest Cretaceous and early Paleocene faunas, so we doubt that there is a strong bias. Additionally, the birth–death diversification analyses account for sampling and preservation biases in the fossil record (see also the comment on that issue above and electronic supplementary material).

Given that this subject was already dealt with in a previous paper (Neubauer et al. 2021, *Commun. Earth Environ.*, <https://doi.org/10.1038/s43247-021-00167-x>) and that the present manuscript focuses on the drivers rather than the actual rates, we would like to avoid getting into detail here.

Line 267: the authors should explain more why no parameter is expected to have a significant contribution in the 20-40 myr bin.

The rates are constant during the 20–40 Myr window (see Fig. 2a in the paper), hence our MBD model aiming to reconstruct these rates through biotic and abiotic factors does not identify any influence of them on diversification. Also considering that the factors vary considerably in that time interval, there can be no significant statistical correlation. We change the sentence in the manuscript to “No factor has a significant contribution in the 20–40 Myr window, which is expected given the constancy of the rates during this time frame compared to variation in the factors (electronic supplementary material, tables S1.1–3)”.

Line 290: for both speciation and extinction? Or just origination? It would be useful to specify here.

Changed as requested.

Reviewer(s)' Comments to Author:

Referee: 1

Comments to the Author(s)

In this manuscript, Neubauer et al. used paleontological models of species diversification to investigate the drivers of diversification of European freshwater gastropods. By focusing on different time windows, they reported that the relative influence of different abiotic and biotic diversification drivers can change through time.

Overall, I think this manuscript is very interesting and timely. It's also well written, although the authors use many acronyms that complexify the reading. I hope my comments will be constructive and helpful.

We thank the reviewer for the positive and encouraging assessment. We took out the abbreviation for effective sampling size (ESS), but we would like to keep abbreviations such as BDS, MBD, MAPE, MCMC and TRI, since they are all typically used in the field. However, we re-defined them in following chapters to facilitate the reading (as was suggested by Referee 1 in the detailed comments below).

I am not an expert on paleontological diversification models but I do have two concerns/questions about how the results could be impacted by (i) the incomplete and heterogenous sampling and (ii) the interdependence across factors.

First, I am wondering how incomplete and heterogenous sampling can bias the results. For instance, both the mean geographic distance the biogeographic separation are two variables that can be largely influenced by sampling bias. Does the model explicitly take that into account? If not, I would suggest some rarefaction/subsampling of the datasets to assess how the conclusions are robust to the heterogeneity and the incompleteness of the fossil records.

Sampling and preservation biases are an inherent problem to the fossil record. The PyRate analyses include estimates of these biases through time, hence the rates are already corrected for sampling/preservation heterogeneity through time. Detailed information was added to the electronic supplementary material.

We also tested a subsampling approach as suggested. Given the extensive computational effort required for the analyses, a comprehensive test for spatial heterogeneity across multiple time windows, replicates, and subsampling sets, would run several months even on a computing cloud. Nonetheless, we tested the effect on the outcomes of the multivariate birth–death analyses by using a geographic subset of the data and the 0–100-Myr window as representative. For each time bin, we randomly subsampled 80% of the data points and re-calculated the average value for each factor; this was repeated ten times. For each of the ten subsamples, the analyses were run for ten replicates to account for uncertainty in the diversification rates. As in the main analyses, median correlation strength and mean shrinkage weights were computed and the values were compared to the original outcomes.

The results show that the values hardly differ from the original results, suggesting that our analyses are robust against spatial heterogeneity in the factors. We included the new results and a short chapter on the matter in the electronic supplementary material.

Second, the authors acknowledge that the “current limitations of our approach involve the interdependence across factors (e.g. basin size change and topographic changes)” (l. 366-367). I think they should at least show the correlations between the different variables tested, and if some of them are strongly correlated, they should rerun the models with only uncorrelated variables to assess how interdependence across factors can influence their results and their conclusions.

The good thing about the multivariate birth–death model we use is that it is relatively immune/robust to collinearity among variables, unlike variable selection in a standard regression analyses. All relationships between factors and rates are inferred in a single multiple regression that uses regularization techniques such as the horseshoe prior to prevent the model from overfitting and to reduce the risk of false positives. The horseshoe prior is essentially a variable selection method that chooses which and how many factors are useful to explain rate variation. If two factors are correlated the horseshoe prior will likely jump from one to the other.

The only issues are that 1) the model might be unable to choose between two or more correlated factors (there is inevitably some identifiability issues if multiple variables follow the same trends) and 2) the signal for any variable might get weaker if it is spread across different collinear factors. Creating more explicit variable selection algorithms (e.g. using indicators as latent parameters) to turn on/off factors is already planned for future projects.

We expanded the relevant section in the Limitation paragraph to make these issues clearer (lines 360–366). Also, we added more detailed information and a figure showing the collinearity among the variables in the electronic supplementary material.

Other comments:

- It's unclear in the abstract if the authors performed additional developments of the multivariate birth-death model or not. In addition, they should explicitly write that they are using paleontological data only.

We slightly adjusted the MBD analysis by enabling time slices. However, this does not change the model itself (i.e. the equations for the relationships between rates and factors and the inference method). We also added information that we use paleontological data to the Abstract.

- l. 43-49: the authors should maybe also refer to species interactions other than “density dependence” (i.e. competition within a clade) and mention that trophic or mutualistic interactions can also drastically affect species diversification.

Changed as requested.

- l. 56-60: True, but it also exists some extensions of these models that allow testing for the effect of a given factor on diversification at a given slice of time only (see Lewitus et al., 2018).

Thanks for the information. We added this example to the following chapter, where such cases are briefly discussed (including similar results from a paper by the same group of authors).

- l. 60-61: “For example, geological processes like continental fragmentation likely vary at a slower pace than biotic interactions and could therefore mask their effects.” It's unclear to me how this can “mask their effect”? Both would be confounded, but I don't see how the effect of continental fragmentation can be unraveled in this case, even if looking at short time windows. This should be clarified.

This comment links to the same issue as addressed already by the Associate Editor – see reply there.

- l. 78-89: I think that this paragraph about the “choice of the taxonomic level” and how to obtain “reasonable estimates of diversification rates” is quite unclear. For instance, why an ideal model taxon for providing reasonable estimates of diversification rates should have representative coverage of different habitat types and lifestyles? Actually, if the taxa are present in different habitat types and lifestyles, these habitat types and lifestyles should indeed have a representative coverage, but not the other way.

True indeed, we rephrased the respective section to make clear that it is not about the variety of habitats/lifestyles but their representative coverage in the fossil record.

- Given that the Multivariate birth-death analyses are central to the paper, their principle could be better explained in the Methods section, and for instance, the details about the parameters of the MCMC runs could be put in the supplementary material.

We added the requested information to the electronic supplementary material.

- I feel they are too many acronyms in the Result section, figure legends, and in the discussion (BDS, MBD, MAPEs...). They should be avoided or at least be redefined every time.

See reply to the first comment of Referee 1 above.

- l. 267: “No parameter has a significant contribution in the 20–40 Myr window, which is expected given the homogeneity of the rates during this time frame”: I don’t see why no factors are expected to have a contribution. Indeed, if the factor driving the species diversification is also constant, it can explain why rates are constant.

This comment links to the same issue as addressed already by the Associate Editor – see reply there.

- l. 340-342: This sentence is quite unclear to me.

We rephrased the sentence to make it better understandable.

References:

Lewitus E, Bittner L, Malviya S, Bowler C, Morlon H. 2018. Clade-specific diversification dynamics of marine diatoms since the Jurassic. *Nature Ecology and Evolution* 2: 1715–1723.

Referee: 2

Comments to the Author(s)

Although there are numerous analyses of drivers of diversification for marine invertebrates, such studies are still remarkably uncommon for terrestrial invertebrate groups with the possible exception of insects. Neubauer et al. attempt to partially rectify this by analyzing diversification dynamics of European freshwater gastropods over the last 100 million years. They do this by applying Bayesian birth-death-samplly analyses (PyRate) to a database of nearly 25K occurrences of over 3K species from nearly 6K localities. Using three related approaches, the authors find very high extinction rates at the end of the Cretaceous followed by elevated origination rates in the early Cenozoic, fairly low diversification through the remainder of the Paleogene, and then elevated volatility in the Neogene with both origination and (to a lesser extent) extinction rates showing multiple pulses.

Given the long history of using marine invertebrates and terrestrial vertebrates for these sorts of analyses, this is a very welcome addition to diversification studies. The analyses employ very modern and sophisticated tools that treats constant rates through time as a Bayesian analog of a null hypothesis, including of a multivariate analysis that allows assessment of the impact of particular extrinsic parameters on the diversification dynamics of freshwater snails. This stands out from what we usually see in studies of drivers of diversification in that the association is directly modeled rather than used as some inflection point (e.g., rates before and after the Paleocene-Eocene Thermal Maximum).

Overall, I think that this is an excellent study. My sole concerns are that the results and discussion occur largely in a vacuum relative to comparable studies. Something that I think is a real selling point of this study is that it is using gastropods: and we have a lot of diversity studies for marine gastropods for the entire Phanerozoic. Quite a few studies examine Meso-Cenozoic gastropod dynamics: for example, I think that studies using data from the Paleobiology Database (e.g., Alroy 2010 *Science* 329:1191) find diversification dynamics among marine gastropods not dissimilar to those shown here. Regardless, gastropods are one of the few groups for which we can do this with both marine and terrestrial fossils, and that makes for interesting contrasts of evolutionary dynamics. Similarly, it might be worth contrasting these findings with those for mammals. We have a lot of studies looking at mammal diversification, including those from the same region over the same timespan (e.g., Finarelli & Liow 2016 *Biol. J. Linn. Soc.* 118:26). In particular, it jumps out to me that volatility among freshwater snails increases during the Neogene: i.e., the time in which grasslands took over much of the planet and radically altered terrestrial ecosystems. We usually focus on what this did to “truly” terrestrial animals; however, grasslands must have affected the ecologies of lacustrine environments, too.

However, these are fairly small points that I would like to see added at the outset and in the discussion: neither merits more than a paragraph.

We are very grateful to the reviewer for his positive evaluation and the constructive criticism. The “vacuum relative to comparable studies” Referee 2 refers to has a simple reason: there are indeed very few studies that address a changing impact of potential abiotic/biotic controls of diversification (and we refer to them in the Introduction). As we already mentioned briefly in the reply to the Editor Comments above, the studies and knowledge on gastropods mentioned by Referee 2 actually refer to a different topic, i.e., diversity. Our study is not concerned with absolute diversity but the rate of change and the factors that underlie this change. This is a very different way of approaching the study of biodiversity, which comes with different assumptions. Nonetheless, we added a sentence specifically mentioning marine gastropods in the section dealing with diversity dependence (lines 296–297).

Moreover, a comparison between freshwater gastropods and terrestrial organisms like mammals or grasses may be misleading in our opinion. Both mammals and grasses occur in very different ecological settings, are likely to respond differently to climatic/topographic/biotic changes and have different evolutionary

histories. The diversification of freshwater snails in the Neogene is much more likely the result of the development of numerous long-lived lakes in Europe, as we discuss in the manuscript (lines 309–320).

It is not that we are reluctant to do the work here, but any of these comparisons or discussions would stand on very weak grounds and we believe the paper would not benefit from this.

I have the additional particular comments.

Lines 62 – 65: “Little is understood about the changing impact of diversification drivers over time, and the available knowledge is based on incompatible methodological approaches (fossils vs. phylogenies, single clades vs. entire faunas, empirical data vs. simulations), geographic scales (single ecosystem vs. global) and species groups.”

I take issue with this statement! For example, a quick Google Scholar search about the relationship between environmental changes and shifting diversification dynamics for the Ordovician Radiations (i.e., “the Great Ordovician Biodiversification Event”) revealed several papers from 2021 alone. I chose that because it is near and dear to my research interests: but study of the potential drivers of marine diversification from throughout the Phanerozoic (and even Ediacaran) really are quite common. Now, what are not common are comparable studies for terrestrial invertebrates, particularly terrestrial invertebrates that are not insects.

Very good point, we changed the beginning of the paragraph to highlight the paucity of information for non-marine invertebrates. However, the sentence mentioned by Referee 2 deals specifically with the changing impact of drivers (not the diversification dynamics per se or the drivers across extended geological time), and there we know comparatively little indeed. The revised version clarifies this issue.

Lines 115-117: We applied the Reversible Jump Markov Chain Monte

Carlo (RJ-MCMC) algorithm using the birth–death model with shifts (BDS), i.e. speciation and extinction rates are allowed to shift across time frames [35].

I have two comments/questions here. First, PyRate should allow sampling rates to vary through time, too. Assuming that is done, then it should be mentioned. Second, it might be worth noting what analyses are (Reversibly) Jumping between. For standard PyRate analyses, that usually would be an N-parameter model to an N-1 or N+1 parameter model (e.g., one more or fewer shifts in diversification or sampling). With the multivariate analysis, I would guess that it includes jumps from N associations between diversification and an extrinsic parameter to N+1 or N-1 such associations. Regardless, both should be stated. (RJMCMC is becoming widespread: but I don’t know that we are to the point where we can assume that readers know what it does in the way that we can probably assume that they know what a principal components or parsimony analysis does.)

This comment partly links to suggestions of the Associate Editor and Referee 1. As requested, we added more information about the methods in the electronic supplementary material. This includes information about the preservation/sampling correction over time (which PyRate does indeed) and details about the RJ-MCMC algorithm. The multivariate analyses do not directly apply the RJ-MCMC algorithm, they use the outcome of the birth–death model (i.e., speciation and extinction times) inferred with that algorithm. We clarified this in the expanded methods in the electronic supplementary material.

Lines 163-164: “We chose ten parameters to test for their potential influence on speciation and extinction rates in each window, including one biotic (diversity) and nine abiotic factors...”

I almost hate to open this can of worms, but did the authors consider biotic parameters such as the diversification of mammal groups or fish groups, or the diversification of grasses? Given that they find diversity dependence (i.e., “logistic” in the paleo literature!) and also an effect of topography, this might not be too important; however, one question that jumped to my mind was whether the diversification of any other taxa commonly found in freshwater habitats or directly affecting them (as grasslands certainly must have done) coincide with changes in freshwater gastropod diversification. One might not be surprised to see increased volatility given some “Red Queen” scenario in which new additions to the ecospace generated arms races with the snails in one capacity or another.

This is a very interesting idea and definitely worth a thought. In the case of freshwater gastropods, however, the situation is difficult. Groups like mammals or even other freshwater taxa like bivalves or ostracods do not compete for the same resources used by gastropods and hardly compete for space. Thus, we would not expect a strong influence. The only species that commonly interact with the snails are certain taxa of fish (as predators). Grasses may have an indirect impact by affecting the nutrients delivered to nearby freshwater ecosystems. Assessing a potential covariation between the diversification of gastropods with fish or grasses is an interesting task for the future, but this would go beyond the scope of the manuscript (as Referee 2 indicates at the end of the review).

Lines 289 – 291: “There is no single parameter that is relevant in all windows, although diversity-dependence shows a significant impact in most.”

That’s cool! It is also, I think, worth noting that diversity-dependence (or, really, richness-dependence) is found in a lot of marine invertebrate studies, including some including gastropods.

We added a sentence specifically highlighting invertebrate studies and gastropods in lines 296–297, where we mention the importance of diversity-dependence across taxa, time and geography.

These are, I think, fairly minor quibbles/comments/questions. The final one in particular might well be the topic of a completely different paper.

Sincerely,

Peter J. Wagner

University of Nebraska, Lincoln

Appendix B

Reply to reviewers

Dear editor,

Thank you very much for the acceptance of our manuscript. We fixed the minor issues as requested by the associate editor and the reviewer. Below we provide a point-by-point reply (in blue).

Kind regards,

Thomas A. Neubauer

Associate Editor's Comments to Author:

The authors have done a nice job with the revisions and have addressed most reviewer concerns. Reviewer 1 re-reviewed the contribution and is happy with the changes, as am I. I have only a few minor editorial comments below.

Line 42: consider changing to 'A major question in evolutionary biology is the processes that drive diversification'

Changed to "A major question in evolutionary biology concerns the processes that drive species diversification".

Line 52: consider changing 'to' to 'on'

Changed as requested.

Line 78: remove 'the' before 'lacustrine diatoms'

Changed as requested.

Line 131: remove the ‘a’

Changed as requested.

Line 142: this is a bit confusingly worded, please consider revising

Changed to “We ran test analyses to assess a reasonable temporal duration of the time windows, in which the complete time interval (0–100 Myr) can be segmented.”

Line 155: consider rewording to “to avoid biasing the results,”

Changed as requested.

Line 234: consider removing ‘the’ before ‘speciation rate’

Changed as requested.

Line 270: I am not sure what is meant by ‘especially’ here

We deleted the word.

Line 296: starting this sentence with ‘this also concerns’ is a bit confusing to me, as I am not sure what is meant by ‘this’

Changed to “These studies involve ...”.

Line 305: remove ‘the’

Changed as requested.

Line 306: place a comma before ‘but with different...’

Changed as requested.

Reviewer(s)' Comments to Author:

Referee: 1

Comments to the Author(s)

The authors have answered all my questions and suggestions and the manuscript appears very good to my eyes.

Just a very minor typo: the references 8 and 28 are the same.

The duplicate was deleted and the remaining references were renumbered.

In addition we made three minor changes:

L 90 – Changed from “... sufficient number of speciation events to be evaluated ...” to “... sufficient number of speciation and extinction events to be evaluated ...”

L 161–167 – This part was previously a bit ambiguously phrased, we changed the formulation to explain the calculation of the MAPEs better.

L 276 (captions of Table 2) – The order of items listed in the last sentence was changed to match the order in the table.